# Can Residential Greenspace Exposure Improve Pain Experience? A Comparison between Physical Visit and Image Viewing

**DOI:** 10.3390/healthcare9070918

**Published:** 2021-07-20

**Authors:** Hansen Li, Xing Zhang, Shilin Bi, Yang Cao, Guodong Zhang

**Affiliations:** 1Institute of Sports Science, College of Physical Education, Southwest University, Chongqing 400715, China; hanson-swu@foxmail.com; 2Department of Basketball and Volleyball, Chengdu Sport University, Chengdu 610041, China; starz-94@foxmail.com; 3National Institute of Education, Nanyang Technological University, Singapore 637616, Singapore; NIE20.BS@E.NTU.EDU.SG; 4Clinical Epidemiology and Biostatistics, School of Medical Sciences, Örebro Universitet, 70182 Örebro, Sweden; 5Unit of Integrative Epidemiology, Institute of Environmental Medicine, Karolinska Institutet, 17177 Stockholm, Sweden

**Keywords:** urban greenspace, residential greenspace, nature exposure, pain, experimental pain

## Abstract

Reducing the burden of pain via greenspace exposure is a rising research topic. However, insufficient evidence has been found in relation to the environmental effect itself. Residential greenspace, as a convenient but limited natural environment for urban dwellers, has benefits and services yet to be discovered. Therefore, the current study recruited 24 young adults to evaluate the effects of physical visit to, or image viewing of, residential greenspace on pain perception and related psychophysiological outcomes, via simulated pain. Pain threshold and tolerance were recorded via the level of pain stimuli, and pain intensity was evaluated using the Visual Analog Scale (VAS). The state scale of the State–Trait Anxiety Inventory (STAI-S) and two adjective pairs were employed to measure the state anxiety and subjective stress, respectively. Meanwhile, heart rate (HR), heart rate variability (HRV), and blood pressure (BP) were measured to investigate physiological responses. Besides, Scenic Beauty Estimation (SBE) was also employed to assess participants’ preference regarding the experimental environments. The results revealed that visiting the greenspace significantly increased the pain threshold and tolerance, while no significant effect was observed for image viewing. On the other hand, no significant difference was observed in pain-related psychophysiological indices between the experimental settings, but significantly negative associations were found between the scores of SBE and subjective stress and state anxiety. In conclusion, the current study brings experimental evidence of improving pain experience via residential greenspace exposure, while the related psychophysiological benefits require further investigation.

## 1. Introduction

Nature exposure is associated with a range of positive impacts on humans and society. Early in 1984, a view of nature was found to accelerate recovery from surgery [1]. Thereafter, the services of nature have been gradually revealed, and visual stimulus was found to play a key role in human–nature interactions [2], which may effectively improve psychophysiological states [3,4,5]. Therefore, considerable efforts have been made to deliver nature stimuli via video or image [6,7], and some visual images of nature were found to elicit similar effects as viewing real nature [8,9]. 

On the other hand, due to rising urban greening and forestry, urban greenspace such as parks and gardens offer urban dwellers easy access to natural settings [10,11]. Therefore, the physical visit is still deemed as the most practical strategy for the general population to receive the benefits of nature. Additionally, according to previous studies, the state of immersion in natural environments is found to induce sophisticated psychophysiological changes. These findings underline the advantage of physical interaction with nature [12,13]. However, the above positive effects were obtained in highly natural environments, such as forests and lakes, and whether a visit to an urban greenspace can elicit similar health benefits is still unclear [12,13]. 

Recently, there has been rising research interest in reducing the burden of pain via greenspace exposure [14]. However, opportunities were mainly proposed based on indirect evidence from neuroscience, physiology, microbiology, and psychology, while experimental evidence is still lacking [14]. As yet, few experiments have investigated the effectiveness of nature-based therapy in reducing the burden of pain. However, these studies used comprehensive interventions, thus the environmental effect itself could not be clearly identified [15,16]. 

Among various urban greenspaces, residential greenspace is the most accessible for urban dwellers, especially for people who live in highly urbanized regions. Urban dwellers only need to go out of their homes to have physical contact with nature. However, unlike forests or urban parks, residential greenspaces are usually incomplete natural environments with smaller areas and higher levels of air pollution and noise. According to previous reports, residential greenspace may buffer the negative effects of urban stressors, but there still remains controversy about the health benefits of residential greenspace exposure, and its function of relieving pain is yet to be investigated [17,18,19,20]. Therefore, we designed this research to evaluate whether a physical visit and image viewing of a residential greenspace can affect pain perception during experimental pain stimulation, and further checked the differences between them in relieving pain and related psychophysiological responses.

We hypothesize that:

(1) Both the physical visit and image viewing of a residential greenspace can result in a higher threshold and tolerance of pain, while reducing pain intensity during pain stimulation;

(2) Both the physical visit and image viewing of a residential greenspace can reduce negative impacts on pain-related psychophysiological outcomes.

(3) The physical visit will be more efficient than image viewing in relieving pain and related negative outcomes.

## 2. Materials and Methods

### 2.1. Experimental Design

A randomized and controlled crossover trial was carried out to investigate the difference in pain perception between the settings. Participants were randomly assigned to three groups and underwent three experimental settings in different orders, counterbalanced using a Latin square plan. The three settings were:

(1) Control: participants sitting in an empty room and looking straight forward;

(2) Greenspace: participants sitting in a greenspace and looking straight forward;

(3) Image: participants sitting in a room and seeing a monitor that displays the image of the greenspace.

### 2.2. Participants

The sample size was calculated using PASS software (v15). Based on our crossover design, the minimum number of subjects required was estimated to be 18 for an α-level of 0.05 and a power of 0.8. According to previously published studies, sample sizes of 6 to 24 have been employed for the 3 × 3 Latin Square plan [21,22,23,24]. To obtain robust results, twenty-four (12 male and 12 female) healthy and painless young adults were recruited from the community, who met the following inclusion criteria: 

(1) absence of any contraindications for electrical stimulation; 

(2) not taking any prescription painkiller; 

(3) absence of drinking habits;

(4) absence of acute and chronic pain.

All participants were informed that they were participating in a study in which they would receive electrical pain stimuli in three conditions (environments), but the experimental design, including site description, visiting order, and study hypothesis, was concealed. They were also informed that they could stop participating at any moment during the experiment at their will. The signed consent form for participating was obtained from each participant before the experiment. The study protocol was approved by the Institute Research Ethics Committee (IREC) of Southwest University, China. The study was conducted in accordance with the Helsinki Declaration and supervised by the IREC. The baseline information of the participants is shown in Table 1. 

### 2.3. Study Setting

Control: an empty room of 100 m^2^ in a laboratory building, equipped with a 100-inch digital monitor (2.26 m × 1.35 m), which was kept off during the experiment (Figure 1a).

Greenspace: the selected greenspace is in the neighborhood of the participants’ apartment, which covers about 1 hectare, and contains grassland, shrubs, and trees (Figure 1b).

Image: the room for image viewing has the same size and settings as the room for the control. The digital monitor (2.26 m × 1.35 m) was used to display the image of the selected greenspace during the experiment (Figure 1c).

The same type of equipment including chairs, monitors, and physiological detectors were used in the three settings to ensure identical testing conditions. The participants were seated in the same spots in the three settings throughout the experiment.

### 2.4. Pain Stimulation

The pain was simulated using a portable electric generator, and the stimuli were delivered to the inner side of the nondominant forearm via two shock patches (3 × 3 cm) at a spacing of 5 cm. Electric pulse (20 Hz, 100 μs width) was generated for pain stimulation, and the current was increased step by step every 3 s, with an increment of 3 mA for each level. During the pain stimulation, the participants were asked to press a button when they felt the slightest pain (from painless feeling to pain), then the level was recorded as the pain threshold. Thereafter, they were also asked to press the button when they could no longer stand the stimulation, then the stimulation was terminated and the highest level was recorded to represent the individual’s pain tolerance. During the pain simulation, the research staff sat behind the participants and out of the participants’ view. They controlled the electric generator according to the participants’ responses. The threshold and pain tolerance of the participants were then recorded by the research staff.

### 2.5. Procedure

Prior to the experiment, the participants met in the laboratory and signed the informed consent form. The tests for the three settings took place from 2:00 to 5:00 p.m. The three groups started the test simultaneously at the different sites. All participants arrived at the experiment sites on their own (about five minutes’ walk), then took a break (15 min) by sitting on chairs and were informed of the detailed testing procedure. The order of questionnaires and measurements for physiological indices is shown in Figure 2. During the stimulation, the participants were required to look forward to the greenspace or the monitor (sitting two meters from the monitor). Every participant was given an independent interview after the test, and was asked if he/she thought there was an environmental effect. 

### 2.6. Measurements

#### 2.6.1. Ambient Data

The data of temperature and noise were collected at the sitting position before each test, using an ambient thermometer (SmartSen-sor-AS817, Smart Sensor Co., Ltd, Hongkong, China) and a smartphone-based noise detection app (Decibel tester v1.2.1, Beijing Buke Century Technology Co., Ltd, Beijing, China; obtained via Huawei app store; ran on a Redmi K30 smartphone), respectively. The thermometer and smartphone were placed on the testing chair and recorded for three minutes, and mean values were eventually adopted. 

#### 2.6.2. Physiological Measurements

Blood pressure is closely related to pain experience, which is regulated by multiple factors such as intensity and duration of pain stimulation [25]. We measured the participants’ systolic blood pressure (SBP) and diastolic blood pressure (DBP) in a relaxing seated position using portable electronic sphygmomanometers (OMRON HEM-7211), and the mean arterial pressure (MAP) was later calculated as ((DBP × 2) + SBP)/3. The blood pressure was measured both before and after the pain stimulation (Figure 2).

The heart rate variability (HRV), a known indicator for challenge and threat states [26], is useful to detect pain responses in inconvenient testing conditions [27]. We ran the HRV4training^TM^ to detect variations in HRV. The application is designed based on photoplethysmography (PPG) and has been validated and applied in studies to investigate physiological response [28,29,30]. The output data include the stand deviation of normal-to-normal intervals (SDNN) and the root mean square of successive R-R intervals (RMSSD) calculated based on a 60 s time frame. The measurement of HRV was performed immediately after the stimulation, and the heart rate (HR) was also recorded while measuring the HRV (Figure 2).

#### 2.6.3. Psychological Measurements

In the present study, several questionnaires were employed to measure the psychological indices (Figure 2), including:

(1) The Scenic Beauty Estimation (SBE), which assesses the subjective preference for environments using a seven-point Likert scale (ranged from −3 to 3 points) and has been widely used in the assessment of scenic beauty for both natural and urban environments [31,32]. In the present study, the SBE was performed in the pre-test, where the participants were asked to sit in the testing position and evaluate the view according to their subjective feeling.

(2) The subjective stress was measured using two adjective pairs, which are obtained from the Short Adjective Check List [33], a widely adopted methodology [34,35]. The adjective pairs are tense–relaxed and nervous–calm. According to Aslaksen and Lyby [36], an 11-point Likert scale (0 to10 points) was used to measure the adjective pairs, and the total stress score was calculated as the mean of the two adjective pairs’ scores. The subjective stress was measured before and after the pain stimulation.

(3) The Visual Analog Scale (VAS), an effective tool to measure the intensity of acute pain, was used to represent from no pain (0 cm) to the imaginably worst pain (10 cm) on a 10 cm line [37]. Participants were asked to mark on the line to represent the most intensive pain that they felt during the stimulation. The measurement was performed immediately after the termination of stimulation. 

(4) A short-form of the state scale of the Spielberger State–Trait Anxiety Inventory (STAI-S), which contains six items to measure state anxiety, was employed to measure the participants’ state anxiety [38]. The shortened form has been used in a number of studies [39,40].

### 2.7. Statistical Analysis

All the data was processed via SPSS 25.0 (SPSS Inc., Chicago, IL, USA). Given the small sample size, the Shapiro-Wilk test was employed to determine the distribution of the variables, and most of the variables turned out to be non-normally distributed. For some data that were difficult to normalize, the Wilcoxon test was employed to check the differences between pre- and post-tests, and the Friedman test was used to check the difference in ambient data between the three settings. The Fisher’s exact test was employed to compare the proportion of the participants who gave responses in the interview in each setting. A two-sided *p*-value < 0.05 was considered statistically significant in the present study, and the post-hoc comparison used a *p*-value adjusted with Bonferroni correction. The generalized liner mixed model (GLMM) was employed to check the environmental effects on pain experience and variations of psychophysiological indices during the pain stimulation, the experimental setting (three levels: control, image viewing and greenspace) was entered as the fixed factor, and the participants were entered as random factor.

## 3. Results

### 3.1. Environmental Characteristics

The ambient temperature and noise were measured during the experiment, and no statistically significant difference was found between the three settings. In terms of the SBE outcomes, the view in the greenspace scored (mean = 1.96, standard deviation (SD) = 0.88) significantly higher than those in the control (mean = −0.33, SD = 0.98, *p* < 0.001) and image viewing (mean = −0.46, SD = 1.13, *p* < 0.001), while no statistically significant change was found between the latter two settings (Table 2). 

### 3.2. Pain Experience in Different Settings

There was no statistically significant difference observed in the baseline psychophysiological outcomes between the three settings. The GLMM demonstrated a statistically significant effect of experimental setting on pain threshold (F = 5.017, *p* = 0.009) and tolerance (F = 11.703, *p* < 0.001). However, no statistically significant effect was observed in pain intensity (VAS score) and any other psychophysiological variations (Table 3). 

The post hoc pairwise comparisons with Bonferroni correction revealed a higher level of pain threshold in greenspace (mean = 5.56, SD = 2.62) than in control (mean = 4.54, SD = 2.27, *p* = 0.018) and image viewing (mean = 4.63, SD = 2.19, *p* = 0.021), while no statistically significant difference was observed between the letter two settings (*p* > 0.05) (Figure 3a). On the other hand, a higher pain tolerance was recorded in greenspace (mean = 11.42, SD = 3.48) than in control (mean = 9.08, SD = 4.02, *p* = 0.001) and image viewing (mean = 8.67, SD = 3.46, *p* < 0.001) (Figure 3b).

According to the interview results, 17% of the participants reported that the control setting was associated with aggravating pain. Meanwhile, 21% reported that image viewing was associated with pain relief, and 50% reported that the greenspace was associated with pain relief (Table 4). The Fisher’s exact test revealed a statistically significant difference in the proportion of participants with personal opinions to pain (*p* = 0.015, Table 4), and the post hoc comparison further revealed a statistically significant difference between the greenspace and control settings (*p* < 0.05).

After removing the data that were associated with personal opinions, the analysis was re-conducted to check the effects in participants who did not expect the potential environmental effects (non-responders). The GLMM demonstrated that the effects on pain threshold became non-significant (F = 1.010, *p* = 0.372), while the effects on pain tolerance remained statistically significant (F = 4.718, *p* = 0.013) (Table 5).

The post hoc pairwise comparisons with Bonferroni correction showed that the pain tolerance was statistically significant higher in greenspace (mean = 10.83, SD = 3.69) than in image viewing (mean = 8.63, SD = 3.48, *p* = 0.014) and control (mean = 8.65, SD = 3.93, *p* = 0.031) (Figure 4). 

### 3.3. Relationships between the Measured Indices

Significant correlations were observed among multiple variables (Figure 5). Specially, the SBE score was significantly correlated with those of subjective stress (Pearson’s correlation coefficient r = −0.308, *p* = 0.008) and state anxiety recorded after the stimulation (r = −0.410, *p* < 0.001, Figure 5). 

## 4. Discussion

In the presented study, we investigated and compared the effects of physical visit and image viewing of a residential greenspace on experimental pain. The overall results demonstrate that the greenspace environment might induce higher pain threshold and tolerance. However, no statistically significant difference was found in the pain-related psychological and physiological outcomes between the experimental settings.

### 4.1. Pain Experience

In the current study, we found that the residential greenspace exposure significantly increased pain threshold and tolerance during the experimental pain, indicating that greenspace exposure may exert positive impacts on pain perception, which partly supports our first hypothesis. According to Stanhope, et al. [14], though current evidence indicates that nature exposure may improve painful conditions, the existing studies have not been designed with appropriate controls to ascertain whether greenspace exposure itself led to the benefits or whether these benefits could be due to involved physical exercise and other activities. Therefore, our study provides evidence of greenspace exposure alone in improving pain experience. Besides, different from studies with nature exposure in total natural forest [41,42], the present study investigated the effects of a small residential greenspace with fewer plants, which partially supported previous research that limited urban greenspace could exert psychological benefits [41].

In the theory of pain, the placebo effect can be triggered by verbal instructions, conditioning, social observation, and interactions [43], which can reduce sensitivity to pain, and thus has been widely applied to cope with pain issues [44,45]. Due to the characteristics of environmental interventions, blinding is hard to carry out for participants [46]. Besides, owing to environmental education and propaganda, the role of nature in health boosting is underlined in public awareness, which may result in a placebo effect in painful conditions [47,48]. Therefore, we interviewed those with personal opinions, and found that up to 50% of participants thought that the greenspace relieved their pain, which implies a potential placebo effect. Although the placebo effect is deemed as a part of nature-based interventions [49], we still checked the results of participants who were not aware of pain relief provided by nature and tried to identify additional effects. As a result, we found that, though the difference in pain intensity became non-significant, the level of pain tolerance was still significantly higher in greenspace, which indicates that greenspace exposure may be more than a placebo. According to Kline [50], the pain relief of greenspace exposure could result from the fact that the natural stimulus may distract the attention from pain perception. In addition, Stanhope, et al. [14] proposed other potential reasons for pain relief in a short-term greenspace exposure. For instance, phytoncides may influence the human immune system, enhance natural killer cells’ activity and treat some types of pain [51,52]. Besides, negative air ions generated by plants may alter pain outcomes through a range of psychophysiological processes [53]. Nevertheless, these mechanisms were proposed based on indirect evidence, and more research is still needed.

In terms of image viewing, no statistically significant difference was found in the level of pain threshold, tolerance, or pain intensity between image viewing and control settings, indicating weak effects of viewing a greenspace image, which does not support our first hypothesis. This null result was similar to those by Lee, et al. [54], who found that a natural visual stimulus did not decrease the dose of sedative medication required for colonoscopy. These results imply the ineffectiveness of nature-based stimuli in some extreme cases. Though distraction effects of nature images have been proved effective to promote attentional resources [55,56], such a level of distraction may not be sufficient to alter intense psychological responses. 

### 4.2. Pain Related Psychophysiological Response

In the present study, HRV, HR, and blood pressure were measured to compare the physiological difference between conditions, whilst state anxiety and subjective stress were measured for the comparison of psychological outcomes. However, no statistically significant difference was found in any of the comparisons, which does not support our second hypothesis. According to our experimental design, self-regulated stimulation was applied to the participants, which means that participants encountered extreme painful stimuli in each of the testing scenarios. Therefore, the similar physiological responses might be due to the similar painful conditions in the three experimental settings. These results indicate that greenspace exposure may not alter psychophysiological responses when suffering pain that is close to an individual’s limit of tolerance.

### 4.3. Physical Visit vs. Image Viewing

The results demonstrated that residential greenspace exposure induced a significantly higher level of pain tolerance than viewing an image of greenspace, which partly supported our third hypothesis. According to the relevant studies, the subtle effects of image viewing may be related to the following factors. First, conventional two-dimensional static media may not be able to duplicate a visual experience of real nature, thus may fail to deliver environmental effects. Second, the benefits of displayed nature may be regulated by the media adopted, and dynamic videos may be better than static pictures in pain relief [6]. Third, the environmental effects are likely to be affected by subjective preference [55,57]. Meidenbauer, et al. [58] argued that the emotional benefits of nature exposure were regulated by subjective preference, which was associated with aesthetic experience. In the current study, the negative association between scenic beauty and negative emotions (stress and anxiety) somewhat supported the above theory. However, due to the display equipment and photography, the image of greenspace was poorly evaluated by the participants, which may be a reason for the subtle effects. In general, our findings suggest that, in addition to the known positive roles of urban parks in nature exposure, residential greenspace may also provide beneficial health resources for urban dwellers, and physical visits are encouraged to maximize these benefits [10,59]. Hence, more stay-in facilities are recommended in the urban greening of neighborhood areas. On the other hand, though the image of greenspace was not effective in the current study, future studies may consider more attractive images of nature, and resort to advanced media technologies to explore solutions using simulated nature exposure for healthcare.

### 4.4. Limitations

The current study aimed to provide evidence that encourages community residents to go outdoors and visit their neighborhood greenspace. Therefore, only the conditions of indoor and greenspace exposure were compared. However, though studies have proved that gray building areas do not contribute to psychophysiological improvements when compared to natural environments, cases for painful conditions were uninvestigated [60,61,62,63]. Future studies may need to use other outdoor controls and find out whether simply getting downstairs can benefit the pain experience. On the other hand, due to our research conditions and the exploratory purpose, we recruited young adults only, which does not allow us to generalize our findings to different populations. Besides, we used portable equipment to carry out the experiment due to the inconvenience of outdoor tests. Though the validity of equipment has been confirmed in previous study, further study may need more sophisticated instruments to obtain more accurate data for quantitative analysis.

## 5. Conclusions

This study investigated the effects of physical visits and image viewing of a residential greenspace on pain perception and related psychophysiological outcomes. Our findings suggest that residential greenspace exposure can enhance pain threshold and tolerance. However, image viewing showed poor effects on pain relief. These findings provide experimental evidence for the positive role of urban greenspace in nature exposure. Based on the findings and limitations of the current study, in future work we will consider other media such as virtual reality (VR) to duplicate the sense of immersion in a natural environment. Furthermore, from the perspective of practical application in healthcare and medical services, future experiments will recruit patients with pain symptoms to re-examine the pain relief effect of residential greenspace exposure.

## Figures and Tables

**Figure 1 healthcare-09-00918-f001:**
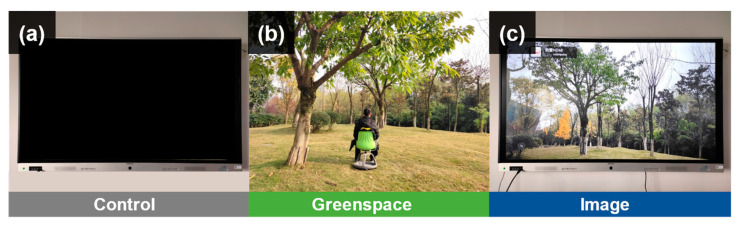
Views of control (**a**), image (**b**), and greenspace (**c**).

**Figure 2 healthcare-09-00918-f002:**
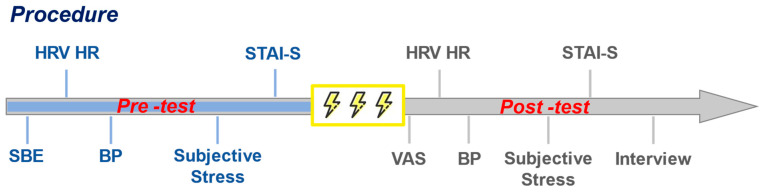
The measurements and testing procedure.

**Figure 3 healthcare-09-00918-f003:**
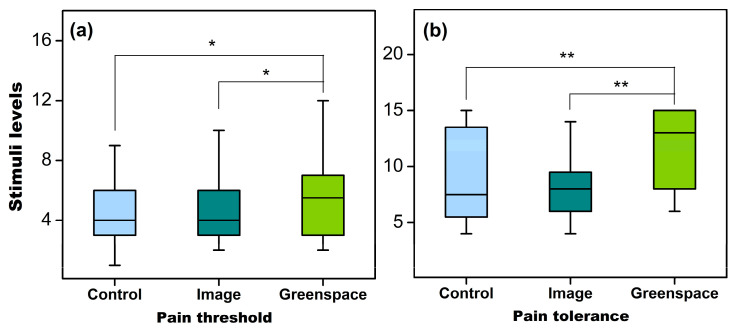
The differences in pain threshold (**a**) and tolerance (**b**) between settings. *, *p* < 0.05; **, *p* < 0.01, adjusted by Bonferroni correction.

**Figure 4 healthcare-09-00918-f004:**
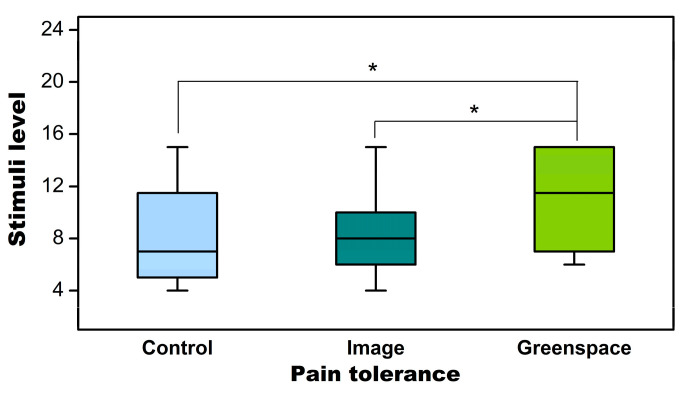
The differences in pain tolerance in non-responders between settings. *, *p* < 0.05, adjusted by Bonferroni correction.

**Figure 5 healthcare-09-00918-f005:**
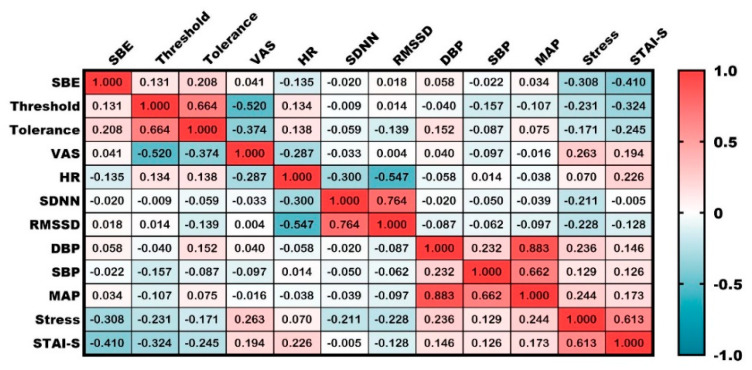
Correlation of measured indices. Numbers in the cells represent Pearson’s r. Analyzed via Pearson’s correlation analysis.

**Table 1 healthcare-09-00918-t001:** Baseline characteristics of the participants.

Gender	*n*	Age (Year)	Height (cm)	Weight (kg)	BMI (kg/m^2^)	Sports Habits (%)
Male	12	21.67 (3.26)	176.67 (7.38)	73.67 (12.23)	23.53 (3.53)	83.33%
Female	12	19.00 (1.04)	162.50 (5.27)	53.50 (7.48)	20.21 (2.18)	66.67%

Note: Data were demonstrated as mean (standard deviation).

**Table 2 healthcare-09-00918-t002:** Environmental parameters and scenic beauty assessment.

Setting	Temperature (°C)	Noise (dB)	SBE (Point)
Control	13.52 (0.45)	40.56 (0.68)	−0.33 (0.98) ^a^
Image	13.53 (0.45)	40.78 (0.78)	−0.46 (1.13) ^a^
Greenspace	13.01 (0.41)	41.00 (1.05)	1.96 (0.88) ^b^

Note: the data were demonstrated as mean (standard error). The different lowercase letters (^a^ or ^b^) indicate statistically significant difference between the settings (*p* < 0.001). Temperature and noise were analyzed via Kruskal-Wallis test; the SBE was analyzed via Friedman test.

**Table 3 healthcare-09-00918-t003:** Fixed effects of setting on the measured indices.

Measurements	F	df 1	df 2	*p*
Pain threshold	5.017	2	69	0.009
Pain tolerance	11.703	2	69	<0.001
VAS	2.529	2	69	0.087
Variation of Subjective stress	0.050	2	69	0.951
Variation of STAI-S	0.817	2	69	0.446
Variation of HR	0.128	2	69	0.880
Variation of SDNN	1.304	2	69	0.278
Variation of RMSSD	1.174	2	69	0.315
Variation of DBP	0.162	2	69	0.851
Variation of SBP	2.035	2	69	0.139
Variation of MAP	0.589	2	69	0.558

**Table 4 healthcare-09-00918-t004:** Differences in proportions of participant with subjective responses.

Effect	Control	Image	Greenspace
Aggravate pain	17% ^a^	0% ^a^	0% ^a^
No effect	83% ^b^	79% ^a,b^	50% ^a^
Relieve pain	0% ^a^	21% ^a,b^	50% ^b^

Note: analyzed via Fisher’s exact test. The different lowercase letters (^a^ or ^b^) indicate significant difference in proportions of each effect (aggravate pain, no effect, or relieve pain) between the control, image, and greenspace settings (*p* < 0.05), using a *p*-value adjusted by Bonferroni correction.

**Table 5 healthcare-09-00918-t005:** Fixed effects of setting on pain threshold and tolerance in non-responders.

Measurements	F	df 1	df 2	*p*
Pain threshold	1.010	2	48	0.372
Pain tolerance	4.718	2	48	0.013

## Data Availability

Data is available on request for corresponding authors.

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
