# Peer review of "Can Residential Greenspace Exposure Improve Pain Experience? A Comparison between Physical Visit and Image Viewing"

_healthcare, 2021, doi:10.3390/healthcare9070918_

Round 1

Reviewer 1 Report

The reviewed manuscript is very interesting a pilot study dealing with "greenspace" exposure and pain experience in young adults. There are some issues that should be carefully thought out:

1. No specific data such as p or r should be given in the abstract. The abstract is generally quite long and convoluted.

2. L61-62 redundant sentence

3. Hypothesis 2 and 3 indicate a broader issue than indicated by the title of the work. I suggest modifying the title.

4. L115: Were they all in the same spot in the field? Did the chairs in the field differ from those in the laboratory (seating comfort)?

5. What do you exactly mean by "greenspace" (L118). In the conclusion and in table 4 you wrote "forest" or "urban forest". Be consistent.

6. L125 How did you use the equipment in your research here?

7. L146 The abundance of abbreviations in this article requires the creation of a legend after the abstract.

8. L204 There was nothing in the methodology about these measurements. Write something, especially what you measured and what accuracy.

9. L212 dB are a logarithmic scale. How did you handle it here?

10. L243: Are the letters written after the test correct here? It is not clear what they refer to.

Reviewer 2 Report

Dear authors,

You have conducted a very interesting study and grafted a very well written accompanying manuscript, which I enjoyed reading.

Your focus on the topic of green space exposure and associated effects on the experience of pain is current and highly policy relevant. I commend you for using quantitative and qualitative data collection methods; you have clearly justified why you have chosen all the methods you are using. Your research design is well thought trough and you have carefully considered strengths and weaknesses of using the chosen design. Your study brings experimental evidence to the area of pain treatment via residential greenspace exposure, and the associated changes to the experience and perception of pain. In addition, you are able to highlight the need for further investigations of related psychophysiological benefits.

I have very few comments for you, as I believe it is a very well thought trough study and a clear and concise manuscript. I believe the English language and style used is excellent and there is only a few places where you should consider reformatting a sentences, or where there perhaps is a typo;

Line 18: ‘most accessible’

Line 27: ‘testing views’

Line 139: ‘by walk’

Line 272: ‘its image’

Line 314: ‘nature visual stimulus of nature’

Line 326: ‘Therefore, the non-significant differences may result from the equivalent perceived stimulations in the different experimental settings.’

Line 333: ‘than its image viewing’

Line 335: ‘2D medium may hard to duplicate’

Line336: ‘the effects of nature stimulus were affected by the means of media, the image is static and may not be as effective as video in pain treatment.’

Line 354: ‘its image’

Line 356: ‘forest bathing’ (this is the first and only time you use this term. For the sake of consistency, I would chose e.g. ‘exposure to urban residential green space’.

Line 357: ‘These findings provide experimental evidence for services and functions of limited urban greenspace.’

I have highlighted this in yellow in the attached file.

You have, at the end of the conclusion, written that further research is warranted. As you are proposing to publish your research as a pilot study, I would suggest that you make it clear what your subsequent plans are. I am assuming the choice to conduct a pilot study was made with the hope/expectations that a larger study is to follow at some point. Also, it would be useful if you included a bit more detail in general about the possible research, which could be conducted following your study, to build on you findings.

As I have very few comments regarding improvements to your manuscript, I have made a list of the main positive points I identified when reading the manuscript and the reason I found it enjoyable to read:

  • English language and style clear and concise all the way through;
  • Very ‘reader-friendly’ all the way through;
  • Abstract gives a clear picture of the content and findings are not exaggerated, selective or omitted;
  • The introduction clearly provides sufficient background information and there is a good use of relevant references;
  • The hypotheses are clearly stated at the beginning;
  • An appropriate research design is chosen and described in adequate detail;
  • I commend the use of mixed methods. All choices related to the methods are well described and justified;
  • Everything is described in such a way that it would be possible for other researchers to repeat the study, thus making it easier for others to build on your findings and benefit from your research;
  • I particularly like the use of photos to illustrated the three different treatment conditions;
  • The results are clearly presented. I particularly like the visualisation of results by using tables and figures, which is underpinned by clear and concise descriptions in the text;
  • The conclusions are underpinned by the results;
  • The initial three hypotheses are revisited in the discussion: it could perhaps be made a bit clearer i.e. addressing them as hypothesis 1, 2, and 3 under a separate heading.

I thoroughly enjoyed reviewing your manuscript and wish you the best of luck with the publication.

Reviewer 3 Report

Thank you for your hard work and showing the significant results in this study. This manuscript has novelty, but can be improved by more explanation for statistics or selecting number of group, and further discussion. 

Line 95-96: Could you add more explanation how to decide the number of sample size for non-statistical expert? It would be better to add references and simple mechanism.

Discussion

  1. Line 302, previous study mentioned that pain relief of greenspace exposure could result from the fact that the natural stimulus may distract the attention for pain perception. I wonder gray infrastructure environment (not greenspace) also can distract the attention for pain perception. But in this study, there is no control experiment setting of other images or other outdoor environment. Please discuss this limitation in the discussion.
  2. 3. Physical visit vs. image viewing. Could you discuss the implication of this result? We understand that there are different pain relief level between real visit and image viewing. But here, which information/implication/policy-related things can be obtained? Please discuss why you want to discern the effect of visiting and viewing image and what can be obtained?
  3. Line 285: Could you compare the different effect of urban green space and forests?

Round 2

Reviewer 3 Report

Thank you for the revising work. I think the manuscript have been modified well by considering reviewers' comments.

line 720: reference 63 doi format is different. 

There are a few references without doi. If you can find the doi, please attach them.